# Spatially Resolved Rainfall Streamflow Modeling in Central Europe

Marc Aurel Vischer<sup>1</sup>, Noelia Otero<sup>1</sup>, and Jackie Ma<sup>1</sup>

<sup>1</sup>Fraunhofer Heinrich-Hertz Institute, Applied Machine Learning Group, 10587 Berlin, Germany

Correspondence: Marc Aurel Vischer (marc.aurel.vischer@hhi.fraunhofer.de) and Jackie Ma (jackie.ma@hhi.fraunhofer.de)

Abstract. Climate change increases the risk of disastrous floods and makes intelligent fresh water management an ever more important issue for society. A central prerequisite is the ability to accurately predict the water level in rivers from a range of predictors, mainly meteorological forecasts. The field of rainfall runoff modeling has seen neural network models surge in popularity over the past few years, but a lot of this early research on model design has been conducted on catchments with smaller size and a low degree of human impact to ensure optimal conditions. Here we present a pipeline that extends the previous neural network approaches in order to better suit the requirements of larger catchments or those characterized by human activity. Unlike previous studies, we do not aggregate the inputs per catchment, but train a neural network to predict local runoff spatially resolved on a regular grid. In a second stage, another neural network routes these quantities into and along entire river networks. The whole pipeline is trained end-to-end, exclusively on empirical data. We show that this architecture is able to capture spatial variation and model large catchments accurately, while increasing data efficiency. Furthermore, it offers the possibility of interpreting and influencing internal states due to its simple design. Our contribution helps to make neural networks more operations-ready in this field and opens up new possibilities to more explicitly account for human activity in the water cycle.

## 1 Introduction

As one of the most frequent and destructive natural disasters, floods are expected to become more common due to climate change (Bevacqua et al., 2021) and more hazardous as the worldwide population in high risk areas is likely to increase (Kam et al., 2021). Heavy precipitation is expected to become more frequent, which will increase flooding risks (Gründemann et al., 2022). Europe is becoming increasingly more vulnerable to flooding due to large-scale atmospheric patterns that lead to widespread precipitation extremes (Bevacqua et al., 2021) and certain landscape properties. This study focuses on river floods in central Europe, where heavy precipitation and snow melt are driving the expansion of flood-impacted areas (Fang et al., 2024). Accurate prediction of such events is the foundation for creating resilience and preventing material damages, displacement of people, and loss of human lives. The field of hydrological research concerned with predicting river levels from meteorological variables is called rainfall streamflow modeling<sup>1</sup>. The aim is to capture the process by which precipitation feeds into rivers and other bodies of water. Predicting runoff, i.e. the amount of excess precipitation being drained away on

<sup>&</sup>lt;sup>1</sup>Another commonly used term is rainfall runoff modeling. As this paper aims to predict streamflow in rivers, we decided to use the more specific term rainfall streamflow modeling, but we will use rainfall runoff modeling to refer to the general literature.

the surface, requires modeling different processes that take place inside or right above the ground, such as evaporation and seepage. It hinges on keeping some record of the state of the surface, e.g. the amount of precipitation in the last days or how much water is stored as snow during winter season. These processes are highly localized, and as a next step, the resulting local runoff needs to be converted into streamflow along a network of rivers. This modeling step is called routing. There is a large body of research that employs (conceptually simplified) physical models for both these tasks (Beven, 2012). Furthermore, models based on neural networks have increasingly been proposed in recent years, e.g. Kratzert et al. (2018); Nearing et al. (2024). We build upon this line of work by introducing a neural network that performs both local rainfall-runoff modeling *on a regular grid* and routing along the river network to predict streamflow time series measured at river gauges. The model is trained to do this in a data-driven, end-to-end fashion. This spatially distributed modeling framework is able to capture spatial co-variability of the input features, thereby enhancing prediction accuracy in larger basins (Yu et al., 2024). It also allows for controllability and scientific discovery, and it is ready to scale to higher spatial and temporal resolution.

The following Section 2 discusses in detail which types of neural networks have been considered for rainfall runoff prediction and routing, and explains our contribution to this ongoing field of research. In Section 3 we introduce a novel, publicly available dataset for spatially resolved rainfall streamflow modeling in five river basins in Germany and neighboring countries, and we describe our model architecture in Section 4. Section 5 presents the main results from the experiments. Section 6 concludes with a brief outlook onto future directions, highlighting the influence of human activity.

#### 2 Related Work

45

We start this section by introducing a classification scheme for rainfall streamflow models. This scheme will provide orientation as we subsequently present previous work on neural networks in rainfall runoff modeling in general, and spatially resolved processing and routing in particular. We carve out how our approach is different and end this section with an overview over this paper's contributions.

# 2.1 Typology of Rainfall Runoff Models

We adapt a classification scheme for rainfall streamflow models originally introduced by Sitterson et al. (2018): Depending on the level of abstraction, models are said to be *empirical* (also *detailed* or *physical*) if they involve physical equations of the involved processes (Horton et al., 2022). *Conceptual* (or physically inspired) models make some substantial simplifications, but still contain (abstract) subsystems or quantities that can be identified with physical entities. Finally, *statistical* models refrain from explicit modeling of anything physical and instead focus on the statistical relationship between inputs and outputs exclusively. Our approach is based on neural networks and falls into the latter category, while most operational models such as LISFLOOD (Van Der Knijff et al., 2010) fall under the conceptual category.

Another criterion for classifying models is the way space is represented in the model: *Lumped* models aggregate all variables (temporal or static) across a station's catchment area before modeling. Not representing the spatial extension of a catchment can be a reasonable modeling assumption for small catchments, but it implies losing the opportunity to model spatial co-variance

within the catchment. An example of the importance of spatial co-variance is the different effect that heavy rainfall might have over a forested area versus on sealed soil. *Distributed* models on the other hand explicitly model local processes, usually on a grid, less commonly on an irregular mesh vector-based, e.g. Hitokoto and Sakuraba (2020); Sun et al. (2022). Most physical or conceptual models fall under the latter category, as the underlying formulas are local and it is straightforward to resolve them on a regular grid for computation. Neural networks in this domain on the other hand started to be developed as lumped models for a combination of historical and technical reasons which we discuss in the next Subsection 2.2.1. In between sits a class of models called *semi-distributed*, where some sort of sub-structure is modeled. Many routing models fall under this category. An example of a neural network based routing model is Nearing et al. (2024), where a network of gauging stations is modeled with high temporal resolution, but not the processes inside each station's catchment area. As we detail below, our model first predicts runoff fully distributed in space, then mapping these runoffs onto the river network in a second stage.

# 2.2 Neural Networks in Rainfall Runoff Modeling

Neural networks have been used for rainfall runoff modeling since the 1990s (Smith and Eli, 1995), but have surged in popularity since Kratzert et al. (2018), when long short term memory (LSTM) layers (Hochreiter and Schmidhuber, 1997) were employed for the first time. Kratzert et al. (2019c) then described the beneficial effects of adding static location information to the meteorological inputs, albeit in an aggregated manner. This type of model has since been demonstrated to predict streamflow more accurately than models not based on neural networks, across a variety of locations and experimental setups (Lees et al., 2021; Mai et al., 2022; Clark et al., 2024). It also transfers more readily to ungauged basins (Kratzert et al., 2019b). Calibrating physical or conceptual rainfall runoff models usually requires hand-crafting ancillary input features to support the meteorological forcing variables, such as catchments' climate type or hydrological signature (see Beven (2012) for an overview). Sometimes, the dataset is first partitioned into hydrologically homogeneous subsets, on which separate parameters are then calibrated (Beven, 2012). Neural networks do not require such human labor and in contrast profit from processing all catchments indiscriminately and with a single model (Kratzert et al., 2024). As we demonstrate in this study, neural networks can be stacked flexibly into a task-specific pipeline and trained end-to-end, without any manual calibration or intermediate steps. They are capable of extracting task-relevant information from a large array of potentially informative, raw static features (Kratzert et al., 2019c). These data sources can include categorical information such as land cover or soil classes, which can not be readily integrated into physical formulas. Neural networks can also leverage entirely new types of input data, such as largescale remote sensing data (Zhu et al., 2023), concentrations of isotopes (Smith et al., 2023) and chemical compounds (Sterle et al., 2024). In addition, research on the explainability of neural networks has been conducted by Kratzert et al. (2019a) and Lees et al. (2022). These studies focused on identifying hydrological quantities and concepts within neural networks. Cheng et al. (2023) used an explainability framework to extracted hydrological signatures from networks in a data driven fashion. Furthermore, Jiang et al. (2022) show that the use of explainability methods can provide a better understanding of the dominant flooding mechanisms across different catchments. Explainability is crucial for reliable operations in real-life applications because it allows for controlling of risk. It also enables scientific discovery (Shen et al., 2018). In summary, LSTM-based models have been firmly established as state of the art in rainfall runoff modeling with a combination of consistently superior performance and addressing the most pressing concerns regarding reliability, even though many questions remain to be answered. Due to their flexibility, they are primary candidates for entirely novel approaches that will become more relevant as climate change gives rise to questions of human influence and multi-factor disasters.

# 2.2.1 Spatially Resolved Processing

Smith and Eli (1995), the first study on neural networks in rainfall runoff modeling, train a simple, non-recurrent neural network on a five-by-five grid of synthetic data as a proof of concept. Another early example of semi-distributed processing is Hu et al. (2007). The authors evaluate the effect of lumping, but use only five rain gauging stations as input instead of a full grid, and a single catchment as a target. Xiang and Demir (2022), unfortunately not peer-reviewed, presents an architecture closely resembling the first stage of our model, which they call GNRRM-TS: Inputs are processed separately on a regular grid before being aggregated using a manually computed flow direction map. Also here the scope is limited to a single station and the only inputs are precipitation and drainage area of each grid cell. Xie et al. (2022) use LSTMs in a gridded fashion to estimate monthly baseflow instead of daily runoff. They also include static information as inputs, but they train their model on hand-selected subgroups of catchments. Muhebwa et al. (2024) propose a nuanced semi-distributed strategy, which instead of aggregating entire catchments, aggregates regions within a catchment that are similarly far upstream. The resulting set of input features for each region group are concatenated and jointly processed by a LSTM model. Hitokoto and Sakuraba (2020) is an interesting example of using an irregular vector-mesh rather than a regular grid. For each node, a conceptual model provides estimates of local runoff that are then aggregated by iteratively simplifying a mesh using a technique that is inspired by particle filters. Once coarsened to 96 nodes, they use a relatively simple four layer fully connected neural network for routing. This approach can be classified as semi-distributed When considering only the portion of the pipeline managed by the neural network, i.e. after the conceptual model's outputs have been coarsened. Sun et al. (2022) also act on an irregular mesh, this time training a graph neural network on the outputs of a conceptual model. After this pre-training, they fine-tune the neural network on streamflow observations - an elegant way to deal with sparse empirical data in this context. However, their study, too, is limited to a single smaller basin in the western United States. Because their model consists of complexly interleaved graph and time convolution layers, they rely on graph coarsening to be able to scale up this approach to another, larger basin. Yu et al. (2024) propose a combination of a LSTM model on the catchment level and a conceptual model for routing.

#### 2.2.2 Routing

Routing refers to modeling the flow of water between gauging stations in a river network. Neural networks have been successfully employed for this task as well. In this context, streamflow at a given station is predicted from the streamflows of upstream stations alone, typically at an hourly resolution. At such high temporal resolution, routing within the river system can ignore slower process like runoff generation or baseflow, and instead focus entirely on the movement of runoff along the river network. Since the stations within a river system can be conceptualized as nodes in a directed acyclic graph, it seems natural to model this data with a graph neural network, although this term is fairly broad (Bronstein et al., 2021). Example of this approach include Moshe et al. (2020), Sit et al. (2021), Sun et al. (2021), Sun et al. (2022), Nevo et al. (2022) and Nearing

et al. (2024) all of which demonstrated excellent performance in this setting. In comparison, the design of the routing stage in our model as detailed below is much more parsimonious in order to give the user more fine-grained control and interpretability.

Another line of research investigates models that act on networks of rain gauges instead of a regular grid of inputs. Such models can be considered semi-distributed as well. The general focus here seems to be on finding suitable architectures for this task, combining self-attention, LSTM, convolution and more complex graph convolution layers. For example, Chen et al. (2023) intricately stack several LSTM layers to represent the river network structure, while Zhou et al. (2023) propose a mixture of self-attention, recurrency and convolution to build a graph neural network for this routing task. Zhu et al. (2023) aggregate remote-sensing rain data within each catchment in a data-driven fashion by training separate convolutional neural networks for every input product. They then concatenate each sub-basin's lumped information with rain gauge data for further routing in a semi-distributed scheme. This convolutional approach amounts to a more sophisticated form of lumping, as it is concerned with unimodal spatial integration of data that will be integrated and temporally modeled only at a later stage. Hu et al. (2024) partially work on gridded data, namely remote-sensing measurements of rainfall. Each sub-basin is aggregated separately using a convolutional LSTM to produce a spatially aggregated timeseries of rainfall in the sub-basin, concatenate it with static information and recent runoff and continue processing this information in a semi-distributed fashion. The crucial difference here is that the convolutional LSTM serves as a data-driven aggregation mechanism for the gridded rainfall input data, but hydrological modeling again takes place in the semi-distributed domain.

# 2.2.3 CAMELS-type datasets

135

150

We have just discussed several studies featuring individual or a few select catchments. Yet the bulk of large-scale rainfall runoff modeling with neural networks has extensively featured the CAMELS dataset or one of its derivatives, based on Newman et al. (2015) and extended to its current form by Addor et al. (2017). It contains meteorological time series and ancillary data for 671 catchments located within the contiguous United States, manually selected for minimal human impact. This implies that the catchments are relatively small, but on the other hand ensures "laboratory conditions" for hydrological modeling. The downside of this is the limited applicability of findings generated with this data to areas of the world where human influence contributes significantly to streamflow, such as central Europe. But as the dataset covers the contiguous US homogeneously, spans a large area, contains many catchments and with a wide variety of different climates, it offers optimal conditions for training neural networks. And so CAMELS rose to popularity together with the neural network approach in rainfall runoff modeling. Since then, similar datasets were introduced to the public that cover other parts of the world: Chile (Alvarez-Garreton et al., 2018), Great Britain (Coxon et al., 2020), Brazil (Chagas et al., 2020), Australia (Fowler et al., 2021), the upper Danube basin (Klingler et al., 2021), France (Delaigue et al., 2022), Switzerland (Höge et al., 2023), Denmark (Liu et al., 2024) and Germany (Loritz et al., 2024).

# 2.3 Contributions

160

165

In this paper, we present three contributions that go beyond what we have discussed so far: We process the data in a spatially resolved manner without prior aggregation, we use a simple routing module that allows for interpretability, and we train this model in an end-to-end fashion on a novel, spatially resolved dataset in central Europe.

Spatially resolved processing takes place in the first or *local* stage of our model, detailed in Subsection 4.1. Its architecture largely follows the one presented in Kratzert et al. (2019b). Instead of using lumped basins as inputs, we apply the same neural network in parallel to each cell of a regular grid of meteorological time series and ancillary inputs. We show that this finer spatial resolution allows us to capture co-variances and provide regularization, especially benefiting larger catchments. This approach is natural for physical or conceptual models that solve local equations at a given rasterization. Yet, no one to our knowledge has applied a neural network directly to the grid of inputs in a way that scales up to entire river basins. The local stage yields a local runoff quantity for each grid cell, exemplarily visualized in the right panel of Figure 6.

The second or *routing* stage, detailed in Subsection 4.2, consists of only two simple network layers without any nonlinearity, efficiently mapping these local runoff quantities onto a river network. Both stages are trained jointly in an end-to-end fashion on the entire dataset, rendering any kind of expert knowledge obsolete. This also means that the model is fitted exclusively on empirical data, enabling scientific discovery from raw data. We show that the river network connectivity graph can be used as inductive bias to constrain the model to reproduce the river's natural layout. This increases data efficiency and allows for better interpretability. We explain how in principle, although this has yet to be shown in practice, the model can be controlled interactively: Extracting or injecting quantities of water can simulate human influence such as industrial, agricultural or hydroelectric energy generation activity, which significantly contributes to streamflow but is independent of the modeled hydrological processes.

Lastly, the lumped datasets discussed above are unsuited for this spatially resolved modeling approach, since it requires both non-spatially-aggregated inputs and streamflow data for entire basins. Hence, for this study we compiled gridded meteorological and static data as well as river streamflow records for five entire basins in central Europe. These basins are characterized by an overall high level of human activity, compared to the CAMELS dataset. The data is publicly available and described in more detail in Vischer et al. (2025).

#### 3 Data

As discussed above, previously released datasets for rainfall streamflow modeling are unsuited for spatially resolved processing, so we compiled a new, publicly available dataset, referenced in the data availability statement. We present the data sources, preprocessing steps and practical aspects in more detail in a separate publication (Vischer et al., 2025). The river discharge data that we use as targets for training as well as the catchment metadata from which we derive the river connectivity is available from the original provider. We provide code that processes and combines it with the input data after manual download.

<sup>&</sup>lt;sup>2</sup>Orography was adapted from the European Space Agency's Copernicus Global 90 m DEM (GLO-90, doi:10.5270/ESA-c5d3d65) © EuroGeographics for the administrative boundaries in panel (a) and (c).

**Figure 1.** Overview of study area, input grid and data types. (a) The study area comprises 5 basins that cover a contiguous area in central Europe<sup>2</sup>. (b) Cells of input grid (orange) for Upper Danube basin. Catchment boundaries (black) are overlaid with corresponding stations (blue), as well as connecting arrows representing the station connectivity network. Cells along catchment boundaries are assigned entirely to the catchment that contains their center point. (c) Visualizations for one example feature of each type of input. Basin outlines (black) and borders of Germany (turquoise) are plotted for reference.

## 3.1 Study Area, Study Period and Resolution

Our study covers five river basins in Germany and parts of neighboring countries: Elbe, Oder, Weser, Rhine as well as the upstream part of the Danube river up to Bratislava (see Figure 1c, right panel). Due to the sparser coverage of gauging stations in the lower reaches of the Danube basin, we decided to focus on the upper reaches where the station network is more homogeneous. Additionally, the placement of river gauging stations varies across countries, as each follows distinct policies for station location. From a machine learning perspective, this results in diverse sampling strategies across the river network. We decided to limit our study to this region so as not to confound the performance of routing with such different sample distributions. The total study area covers a contiguous 570.581 km² area. Figure 1c also visualizes exemplary features in the study area with boundaries of the river basins and Germany for geographic reference. Based on the consistent availability of streamflow data, we decided to conduct our experiments on the water years 1981-2011. A water year lasts from October 1st of the previous year

to September 30th. Due to data availability, we homogenized all input data to daily temporal resolution and regular grid that is compatible with the ERA5-Land dataset (Muñoz Sabater, 2019) and covers the earth's surface with a spatial resolution of  $0.1^{\circ} \times 0.1^{\circ}$  or roughly  $9km \times 9km$ . If a grid cell is located along a catchment boundary, we assign it entirely to the catchment that contains the cell's center point. This avoids having to represent fractional cells in the pipeline and seemed an acceptable trade-off for the sake of simplicity, considering that the area covered by each grid cell is relatively small.

#### 3.2 Dynamic Input Data

Runoff is primarily driven by precipitation, but temperature and solar radiation need to be taken into account as well to properly capture processes like evaporation or snow dynamics. Thus, our metrological input variables or *forcings* are daily minimum, average and maximum temperature, daily sum and standard deviation of precipitation and average potential evaporation - a score computed from radiation, temperature, air pressure and humidity. This set of variables is widely used in previous studies (Kratzert et al., 2018, 2019b) and was retrieved from ERA5-Land (Muñoz Sabater, 2019). The variables, downloaded at three hour intervals, were aggregated to a daily time step to match the time resolution of the target time series.<sup>3</sup> (Copernicus Climate Change Service, 2022). We amend these six meteorological input features with two more sine-cosine embeddings of day of the week and day of the year, which can be considered as a coarse proxy to human activity (Otero et al., 2023).

## 3.3 Static Input Data

Following the insights from Kratzert et al. (2019c) and Shalev et al. (2019), we include static data, also called ancillary data with the aim of training models jointly on all locations and thus increasing transfer performance. Specifically, we include hydrogeological properties, soil class, land cover, and orographic features derived from a digital elevation map for a total of 46 feature dimensions. Please refer to Vischer et al. (2025) for a detailed description of the origin and preprocessing steps of all input features. The number of inputs could be streamlined in the future, as Cheng et al. (2023) shows how relevance propagation identifies non-task relevant features.

## 3.4 Target Streamflow Time Series, Station Information and River Networks

Target time series of streamflow at each station were obtained from the Global Runoff Data Center (GRDC) data portal. Together with the streamflow data, the GRDC offers a catalog of station information. We considered all stations in the station catalog, but excluded stations that had ten or more values missing in the time series for the selected study period. Furthermore, initial experiments showed that including stations with less than 500 km<sup>2</sup> drainage area in training decreased performance, even when evaluating exclusively on larger stations. We decided to exclude these small catchments, and discuss this decision in Section 6. Another natural limitation on the spatial scope of this approach is that it only captures rainfall-streamflow dynamics in locations contained in the drainage area of a gauging station. In coastal areas, runoff might directly enter the sea through

<sup>&</sup>lt;sup>3</sup>The dataset was downloaded from the Copernicus Climate Change Service (2022). The results contain modified Copernicus Climate Change Service information 2020. Neither the European Commission nor ECMWF is responsible for any use that may be made of the Copernicus information or data it contains.

smaller streams that are not gauged. Hence, our study area usually starts several kilometers inland from the sea. The following number of stations resulted in each basin: 62 in upper Danube, 34 in Elbe, 36 in Oder, 78 in Rhine and 29 in Weser basins, for a total of 239 stations. For comparison, the CAMELS dataset contains 671 catchments. Further visualizations of the river networks can be inspected in the preprocessing scripts, along with all details on how the metadata was processed.

#### 4 Methods

This section provides detail on our model's local and routing stage. This is followed by an explanation of how the river network connectivity graph is calculated and integrated into the routing stage of the model as inductive bias. We then describe the end-to-end training process and which metrics and baselines were used in our experiments.

# 4.1 Model Local Stage - Modeling Spatially Resolved Runoff

**Figure 2.** Overview of the local stage of the network architecture, acting in parallel on each of the input grid cells. Static inputs are reduced to ten features by feeding them through a simple, fully connected embedding layer and applying a nonlinearity function. The resulting feature vector is repeated for every time step **T** of the meteorological forcing time series and concatenated with the time series. The resulting 20 features are fed through a 250-unit LSTM layer. The recurrent layer's output is then reduced to a 1D output time series by sequentially applying two fully connected readout layers plus a nonlinearity in between. Numbers in parentheses signify feature vectors at a given stage in the model pipeline, numbers without parentheses signify the size of a layer's weight matrix.

The task of the model's local stage is to integrate all input modalities and predict local runoff quantities on a regular grid. These local quantities are then fed through the routing layer described in the following subsection. As mentioned before, we follow Kratzert et al. (2019b) by adding static information to the meteorological input and adapting their original model design for the local stage. While this model was originally used on aggregated catchment time series, we apply it to all grid locations in parallel. Also contrary to the original design, we add a simple, fully connected layer that reduces the 46 static input dimensions down to ten dimensions. They are concatenated with another ten dimensions of meteorological forcings for a total

input dimensionality to the LSTM layer of 20, compared to 32 in the case of Kratzert et al. (2019b). Their LSTM layer consists of 256 units, ours of 250. However, we reduce the 250 output values of the LSTM layer by using two regression layers instead of one. We do not employ a nonlinearity after the second readout layer, meaning that the network's outputs are not confined to the range of e.g. [-1,1], but rather live in the range of actual, physical quantities. Figure 2 visualizes the simple network architecture. In summary, the routing stage of our model is largely identical to the model used in Kratzert et al. (2019b) with a few small modifications. We want to emphasize here that the model does not have any predictive capabilities in itself. It uses meteorological forecasts to produce a forecast of runoffs.

In an exploratory experiment, we compared concatenating static inputs to dynamic inputs before and after the LSTM layer. Feeding the static inputs through the LSTM together with the dynamic inputs resulted in substantially better performance. This is consistent with the findings of Kratzert et al. (2019b) and can be explained by the static inputs helping the LSTM to better adapt to the hydrological dynamics of a location. It is similar to training separate models for different climatic zones, but in a data-driven fashion. Indeed, Cheng et al. (2023) show that clustering on the relevance values of the different inputs results in hydrologically plausible clusters. Concatenating all features before feeding them through the LSTM layer requires more parameters in this layer, which due to its intricate inner workings is particularly expensive to train in terms of data and compute. We also used gated recurrent units (GRU) (Cho et al., 2014) instead of LSTMs as a backend, which mitigates this problem a little because they are computationally more efficient. We found that they do not affect performance, but decided to stick with LSTM as our main backbone as it is more common in the literature. Nevertheless, using GRUs could be another way to further optimize the model.

# 4.2 Model Routing Stage - Integrating Local Runoff and Routing

The task of the routing stage of our model is to map the locally generated runoff to a station's catchment area, and then routing the runoff along the river network to predict streamflow time series for every station in the basin. Figure 3 visualizes the layout. Within a given river basin, we concatenate the predicted runoff time series of all grid cells. The network learns a simple, strictly linear mapping consisting of two layers: First, a fully connected layer without nonlinearity maps all grid cells G to their respective stations S. Since no nonlinearity is added, this layer can be translated into a weighted, time dependent average of all grid cells within a catchment. Location information from the station catalog, described below, can be used as an inductive bias to constrain this layer so as to only route water in a physically plausible way. Then, a 1D-convolution layer (Kiranyaz et al., 2021) performs time convolution on each station's time series to combine information inside the river network over the last nine days. Separate kernel values can be learned for each day, but the same kernel is applied jointly over the entire time series. The kernel length of nine days was chosen as a conservative estimate of the maximum time that it would take water to run along the entirety of any of the river networks considered in this study, but of course awaits empirical validation and further optimization. The kernels in this layer are constrained by the connectivity of the river network in order to be physically plausible as we explain in the following subsection. Crucially, this stage does not involve any nonlinearity. Hence, both the fully connected as well as the time convolution layer are linear functions and as such can be chained to form yet another linear function. This means that the quantities of water "flowing" through this pipeline are physically interpretable. Put differently,

**Figure 3.** Overview of the routing stage of the network architecture, acting on the output of the local stage. The local runoff time series of all grid cells in a given basin are concatenated. Together, they are fed through a fully connected layer that projects them down onto the number of the basin's stations. The weight matrix in this layer is constrained to be non-zero only if a grid cell lies in the catchment of a given station. The stations' time series are then time-convolved with a kernel length of 9 time steps to model he flow of water in between stations. Again, each kernel is constrained to be only non-zero only if a given station is directly upstream of another station. The routing stage yields time series for all stations in the basin. It does not include any nonlinearity, so all activation values can be interpreted as streamflow quantities. The basin map icon and dotted arrows indicate that the river connectivity information serves as inductive bias on these two layers, constraining the activations to replicate the real river network.

input quantities can be added or subtracted meaningfully to and from the input. A practical application example of this, which we plan to further investigate, is the injection or extraction of water in between two stations to simulate agricultural, industrial or hydroelectric human activity. This sets our approach apart from previous routing approaches. We also investigated the effect of not constraining the weight matrices with the connectivity matrices, which leads to slightly poorer performance when data is scarce (see Subsections 4.5 and 5.5).

## 4.3 Structural Bias

The station catalog contains polygons of each station's catchment area. From this, we can derive two important kinds of information: First, for every grid cell we can determine in which station's catchment it is located. Second, for each station we can determine how it is connected to upstream and downstream stations. This subsetion explains how this information is extracted and how we use it as structural inductive bias to constrict the routing layer to only consider physically plausible routes of water flow.

#### 4.3.1 Catchment Matrices

Mapping grid cells to stations is important to ensure that the runoff predicted at a given location ends up at the only station that is physically plausible. Since a given station's catchment area is contained within all the downstream stations' catchment areas, we need to make sure that we select the one where the generated runoff first enters the river network. To do so, we select the station with the smallest catchment area that contains a given grid cell. For each river network, we represent this information conveniently in a one-hot matrix with grid cells as rows and stations as columns. This matrix is then used to

constrain the fully connected layer in the routing stage of the network. This is achieved by multiplying the one-hot matrix point-wise with the freshly initialized weight matrix of this layer before training begins. Weights corresponding to physically impossible connections are thus set to zero from the start. Zero weights can not contribute to gradients, and will remain zero throughout the training. All other weights are free to be optimized.

## 4.3.2 Connectivity Matrices

From the catchment area polygons, a graph representing the connectivity between stations in the river network can be derived. Each node in the graph represents a station, a directed edge exists between a station A and B if A is directly upstream of B. We determine this by verifying if the catchment area of A is contained in the catchment area of B, ensuring that there are no intermediary stations in between, i.e. contained by B and containing A. Note that this automatically leads to a directed, acyclic graph. This fits our approach well, as it does not require us to apply any model of routing iteratively in order to capture cyclic movements within the graph. The graph is represented by a connectivity matrix, i.e. a square matrix with rows (input) and columns (output) corresponding to stations, where the entries are 1 if a directed edge exists and 0 otherwise. This matrix is used to constrain the time convolution layer in a manner similar to the catchment matrix. After initialization of the weight matrix, the connectivity matrix is multiplied point-wise, preserving the weights where a connection exists and setting them to zero where no connection is present. The only difference is that the connectivity matrix needs to be repeated by the depth of the temporal convolution, nine times in our case for the nine days of past information that we convolve.

# 4.4 Training and Metrics

310

We split the data into three parts, all containing entire water years: A training set from water years 1981 to 2005, a validation set for model selection from 2005 to 2008, and a test set to report the final performance from 2008 to 2011. We also created two special training datasets to illustrate how the models perform on less training data: A medium length training set ranging from 1991 to 2005 and a short training set from 1999 to 2005. Regardless of the length of the training set, we divide it into chunks of 400 days that partially overlap. The first 30 days are used as a warm up period for the LSTM. During this time no gradients are computed and the LSTM can stabilize into an operating regime before starting the learning process. The value of 400 was chosen to accommodate an entire year plus the warm up period, while still being able to fit the gradients claculated during training into GPU memory. It would not be detrimental to use even longer time series and the model is capable of processing time series of arbitrary length. In fact, no gradients are computed during inference, which significantly reduces the memory footprint and allows us us to calculate all metrics on uninterrupted time series in a single model forward pass. In all experiments, we trained for 2000 epochs of the training data. This number of epochs is generous for all models to converge. Since the purpose of this study is to provide a proof of concept of this type of spatially resolved processing, we decided to not conduct an extensive hyperparameter optimization or perform input feature ablation. Instead, we included all potentially relevant static data and ran hyperparameter tuning experiments only on a limited set of values for a few key hyperparameters, summarized in Table A2. We trained ten different random seeds for every setting, and report the best seed in terms of stationwise median Nash-Sutcliffe Efficiency (NSE) score on the test period.

We use the widespread NSE metric (Nash and Sutcliffe, 1970) both as a loss function in training as well as a score to quantify performance. The NSE normalizes the square loss of each station by the standard deviation of the station's values in the training period, so as to count each station equitably towards the loss or performance, regardless of the magnitude of the river at this point. We do not split training and test partitions geographically, as breaking up basins would make routing impossible. We also refer to Klotz et al. (2024) for a word of caution when combining NSE values that are calculated on partitions of a dataset. Unless noted otherwise, the scores we report were calculated on the test period. The median NSE over all stations serves as a robust point estimate of performance, but for the interested reader we provide mean NSE scores in Table A1, which we found to correlate strongly with the median NSE for all seeds. Likewise, we report the percentage of stations with a NSE score below zero, which indicates predictive performance worse than simply using the average value of a station's runoff for prediction, i.e. chance level. The popular Kling-Gupta efficiency (KGE) metric (Gupta et al., 2009) was developed in the context of univariate, convex optimization - both these assumptions do not hold in the case of training a neural network. However for the sake of comparability, we also report the KGE values in the appendix as well.

As stated before, in this study we refrained from extensive hyperparameter optimization to maximize the performance. A few exploratory experiments to calibrate our pipeline seemed necessary nonetheless: Dropout (0, 10%, 30%, 50% separately in recurrent and readout layers of the routing stage) did not increase performance, so we removed it entirely. 250 units in the LSTM layer (out of 150, 200, 250 and 300) yielded the best results. We use an automatic learning rate scheduler, the ReduceLROnPlateau scheduler provided by Pytorch (Ansel et al., 2024) with threshold 1e-3 and patience 10, so the pipeline trains robustly with regard to the initial learning rate (1e-4, 5e-4, 1e-5). But as the baselines in the experiments have vastly different number of parameters, we decided to continue the experiments by always trying out both of the lower values. Unless explicitly mentioned, we report the performance resulting from the better value for every condition, table A2 lists the results in detail.

## 4.5 Baselines

We introduce two baselines in order to evaluate the performance gains of the two central aspects of our pipeline: spatially resolved processing in the local stage and the inclusion of structural bias in the routing stage. In the first baseline experiment, we aggregate all spatial information within a catchment and feed it through the same architecture as used in the local stage. Normally, this stage of the model processes individual grid cells. But here it processes entire, aggregated catchments, and the output is a prediction of the runoff measured at the corresponding station. The aggregated baseline does not require any further routing. This model and processing pipeline is identical to the one in Kratzert et al. (2019b), apart from the small differences in parameter values listed in Subsection 4.1. As we discussed above, it has been widely used in the literature since its introduction, which allows us to compare our own results from our custom dataset to a wider body of literature. We will call this baseline aggregated, whereas our default model will be referred to as *spatially resolved*.

The second baseline leaves the local stage unaltered, but does not constrain the weight matrices in the routing stage. Instead, we use two fully connected time convolution layers with a kernel size of nine and three days, respectively, and a nonlinearity

in between. The design is intended to be more simple and conventional for neural networks. We will refer to this baseline as *naive routing*, and to our default model as *structured routing*.

## 5 Results and Discussion

We start this section by presenting and contextualizing the general performance of our model. We then show that our model excels in modeling large catchments, is less prone to overfitting and learns from the data more efficiently than the baselines. Inductive bias does not make a big difference to performance. We end this section by showcasing that our model is, unlike most neural networks, not an entirely black box model, and that capturing human influence seems to be the biggest challenge in our study area.

## 5.1 Model Performance

The aggregated baseline achieves a median NSE test performance of 0.69 on our dataset. As discussed above, it is virtually identical to the model from Kratzert et al. (2019b), who report a median NSE of 0.74 on the CAMELS dataset. These results were subsequently confirmed, small modifications of the model parameters or test period notwithstanding, by Shalev et al. (2019) (median NSE 0.73), Acuña Espinoza et al. (2024) (median NSE 0.75) and Frame et al. (2022) (median NSE between 0.72 and 0.81) We conjecture that our baseline performance is probably lower due to our dataset being smaller than CAMELS and likely containing greater human activity. Turning to other regions, the same model was used for example by Mai et al. (2022) for the Great Lakes Area (median KGE of 0.76), by Lees et al. (2021) for Great Britain (median NSE 0.88) and by Loritz et al. (2024) for Germany (median NSE 0.84). The latter two results go against our conjecture for inferior performance due to human signal. The fact that Loritz et al. (2024) use an ensemble of models might be a factor at play, but in any case this requires further investigation. These studies provide valuable comparisons with other model types: In all studies just mentioned, the authors compared the neural network model against a variety of physical, conceptual or hybrid models and consistently found that the alternatives were outperformed by the neural network.

Our model achieves a median NSE performance on the test dataset of 0.77 compared to the baseline of 0.69 and thus appears to be able to compensate somewhat for the more demanding modeling context. Only training and testing our model on a spatially extended version of CAMELS would allow for more direct comparability with other approaches. Unfortunately this has to be left for future research as it requires creating a spatially resolved version of CAMELS first. In any case, despite the data being different and not allowing for a straightforward comparison, this shows that runoff generation and routing can be learned end-to-end by a single model pipeline and without additional data along the way. Moreover, the simplicity of the routing module's design, along with the possibilities it offers, does not come at a significant performance cost. An example in case is that the internal activations inside the network between local and routing stage appear to be hydrologically plausible, as Figure 6 illustrates. We want to stress that this is an emergent property and was not incentivized during training.

| Input Processing   | Routing    | median NSE | NSE 

**Figure 4.** Effect of catchment size: Each point corresponds to a station's performance vs. the size of its catchment, for aggregated baseline (blue) and spatially resolved pipeline (red) on training (left panel) and test (right panel) period. The individual data points are fitted linearly to show the trend. Aggregated processing impairs performance, especially in larger catchments. The trend is more pronounced on the test dataset, indicating overfitting on the training dataset. Spatially resolved processing is less prone to overfitting and manages to handle large catchments accurately in a low data setting.

# 5.3 Inherent Regularization

A comparison between the left and right panel of Figure 4 reveals a substantial performance drop between train and test period. To a certain degree, this is normally to be expected due to overfitting. But the effect is much more pronounced for the aggregated than for the spatially resolved model. It stands to reason that the neural network in the aggregated baseline with its associated reduction of data overfits severely. This also becomes apparent when looking at the median NSE values in training and test datasets. While overall performance drops from 0.86 in training to 0.69 in test for aggregated processing, spatially resolved processing deteriorates more gracefully from 0.90 to 0.77. Spatially resolved processing, with its shared local stage and overall much more data, seems to have an intrinsic regularization effect.

## 5.4 Data Efficiency

405

The positive effect of spatially resolved training, especially for large catchments, becomes even more pronounced when looking at modeling in an environment with limited available training data. Figure 5 visualizes the difference in NSE score between spatial and aggregated processing on a per-catchment level, when training on 25, 15 and six years of training data. Differences are positive across all sizes of catchments, meaning spatial processing on average performs better regardless of the catchment size. Yet the positive trend becomes stronger as data becomes more scarce. Spatially resolved processing utilizes the available data more efficiently.

**Figure 5.** Effect of training period length: Each point corresponds to a station's NSE test performance after training for 25 years (green), 15 years (purple), and 6 years (orange). Across all catchment sizes and different amounts of training data, the spatial pipeline (left panel) outperforms the aggregated baseline (right panel). The latter's performance deteriorates quickly when training data is scarce. The spatially resolved pipeline is much more robust in this regard, demonstrating its increased data efficiency.

## 5.5 Inductive Bias

Including inductive bias for what we call structured routing leads to slightly better performance than naive routing without the additional real-world information, resulting in a median NSE of 0.77 compared to 0.72. Figure A1 contains more detailed results, but also shows that the performance gain is small. The point we want to make here is that the practical benefit of being able to simulate the injection or extraction of quantities of water in routing process does not come at the cost of lowering performance. As we mentioned before, naive routing on the other hand does remain an important tool in modeling basins where catchment delineation information is unavailable or unreliable, or where lateral transport of water inside the bedrock layer across catchment boundaries is suspected. Whether or not we use inductive bias in the routing layer, our networks are extremely simple compared to other networks proposed in the literature for routing that we discusses above. Certainly, we demonstrate that routing modules do not need to be complex, and river network extraction algorithms are not necessary for end-to-end routing, e.g. when no catchment boundary information is available.

## 420 5.6 Interpretable Internal States

Internal states of neural networks are usually not interpretable. In our case, however, the activations of the final layer of the local stage which would go into the routing module stage of the model appear to be hydrologically plausible. As an illustrating example, Figure 6 displays these activation values for two exemplary days in spring and summer. The spatio-temporal correlation seems to suggest that in the example day in spring, runoff is primarily driven by snow melt in low mountain ranges, whereas in summer, it seems to be driven by heavy precipitation events. We want to make the point that we did not enforce this property during the training process, e.g. by providing additional target information or training a special readout layer. This is a purely data-driven, emergent behavior, resulting from both the end-to-end training process and the model's parsimonious design. Particularly, we attribute this phenomenon to the shared recurrent layer in the local stage and the combination of inductive bias and linearity of the routing stage. Unlike more complex neural network designs which are generally considered black box models, this suggests that our model naturally allows for a certain degree of internal control by manipulating these internal states, e.g. by subtracting or adding quantities of water to the natural runoff. It also creates new opportunity for further scientific discovery from large quantities of data.

## 5.7 On Human Influence

We conclude this section by discussing a specific negative outlier in terms of station-wise NSE. As explained above, a negative NSE values indicates performance below chance level. The only two stations that yield negative NSE values after training consistently across all ten random seeds - are Spremberg and Boxberg (GRDC numbers 6340800 and 6340810) located along small rivers next to large brown coal surface mining operations in the Lusatia region. A potential explanation for these extreme outliers is that those mining operations have an influence on the overall water balance that is relatively large compared to the hydrological processes in such catchments. This seems to support our assumption that human influence is one of the main obstacles to be overcome by rainfall streamflow models in the densely populated areas of central Europe. We want to emphasize

Figure 6. Internal states of the model are hydrologically interpretable. This figures shows precipitation data from the input (left column) and activation values after the local stage of the model (right column) for two days in March (top row) and August 2006 (bottom row). The days were manually selected for illustration purposes. Values are displayed in arbitrary units, with blue signifying more runoff or precipitation. We included the outline of Germany (© EuroGeographics for the administrative boundaries) in dotted lines for geographic reference. For the day in spring (top row), a low spatial correlation between precipitation and runoff together with a pattern of high runoff values in low mountain ranges suggests that runoff on this day is primarily driven by snow melt in intermediate altitudes. For the day in summer (bottom row) on the other hand, we see runoff that is driven by two clusters of heavy precipitation in the eastern part of our study area.

that the two stations in question only performed very poorly in the validation period, but not in the test period. We hypothesize that operations might have changed in the meantime or the test period simply lacks substantial events of human influence by chance. We plan to investigate this phenomenon more closely and explore potential solutions in future work.

# 6 Summary and Future Directions

We have successfully trained a neural network in an end-to-end fashion to capture runoff generation in a spatially resolved manner on large scale. Training on five entire river basins in central Europe, we have shown that this approach is advantageous especially in large catchments. The parsimonious network design of the module that performs routing within these river basins

is also noteworthy. Not only does this approach mitigate overfitting and increase data efficiency, but the simplicity of the design and the ability to integrate inductive bias opens new possibilities to control the inner workings of the model. This is not common for neural networks.

Our model reaches a level of performance comparable to that of other benchmark models, both conceptual and statistical. In future work, we plan to compare our model to other state-of-the-art models used in science and operations. The creation of a spatially resolved version of the popular CAMELS dataset would make our model's performance directly comparable with a large portion of the neural networks literature. A direct comparison of our overall pipeline to that of an operational system like LISFLOOD would be similarly interesting. Another question that needs to be addressed in the future is how the performance of our model decreases with the forecast horizon. As we discussed before, our model does not have predictive capabilities of its own. Instead, it relies on a suitable meteorological forecast as input to generate a forecast of streamflow quantities, and the quality of the model's predictions thus depends on the quality of the forecast meteorological input. Quantifying this effect is important for real-world applications.

In this study, we excluded catchments where we suspected that human influence is too strong based on a simple catchment area heuristic. Unlike much research in this area, we do not train our model exclusively on catchments with little human influence. But a more sophisticated strategy, inspired e.g. by Loritz et al. (2024) or Tursun et al. (2024) is needed to properly quantify human impact and thus being able to disentangle the effects of catchment area and human influence on performance. Likewise, while we included day of the week and day of the year as simple approximations to human influence, the effectiveness of this measure can only be properly evaluated with a suitable ground truth.

Another big limitation in terms of data availability is the temporal and spatial resolution: The relatively small size of the recurrent layer enables our model to process time series at a higher than daily temporal resolution. Because the local stage is applied in parallel to all input locations, the number of parameters is independent of the number of inputs. This is where most of the computation happens, and its computational demands grow only linearly with the number of locations. The weight matrices in the routing stage grow quadratically, but are much smaller in the first place.

Another important aspect that we will address in future work is discussed in Klotz et al. (2022), where the authors extend a model similar to ours by using the outputs as parameters of a distribution. Such distributional predictions could be obtained from our model in the same way and would be of great interest for many real-world applications. Producing genuinely probabilistic forecasts and warnings in this fashion is theoretically more sound than training an ensemble of more or less different models and combining their predictions.

As would be expected from the high degree of human activity in our study area, we found evidence that the effect of human influence is the central obstacle to further improving model performance in such an environment. Yet, a comprehensive investigation of the extent and impact of this phenomenon is still required. Future research could demonstrate that our neural network is capable of incorporating simulated human activity, such as water extraction or diversion, into the modeling of hydrological processes.

Code and data availability. The data used in this study is publicly available under CC BY-NC-SA license at hydroshare, the code used to preprocess it is available under Clear BSD license at our repository.

**Figure A1.** Station-wise differences in NSE performance between structured and naive routing, plotted against the size of the station's catchment on training (left panel) and test period (right panel). Positive values indicate that structured routing performs better. The individual data points are fitted linearly. Structured routing marginally outperforms naive routing on the test dataset (indicated by values greater zero). On the training dataset, naive routing performs better when data is scarce (values below zero for short training period in green), indicating overfitting.

| Input Processing | Routing    | Train Period | median NSE | mean NSE | NSE < 0 | median KGE | mean KGE |
|------------------|------------|--------------|------------|----------|---------|------------|----------|
| spatially res.   | structured | long         | 0.773      | 0.751    | 0.000   | 0.803      | 0.779    |
| spatially res.   | naive      | long         | 0.719      | 0.706    | 0.000   | 0.791      | 0.775    |
| aggregated       | -          | long         | 0.691      | 0.643    | 0.013   | 0.731      | 0.702    |
| spatially res.   | structured | med.         | 0.739      | 0.717    | 0.004   | 0.775      | 0.753    |
| spatially res.   | naive      | med.         | 0.735      | 0.692    | 0.008   | 0.776      | 0.744    |
| aggregated       | -          | med.         | 0.642      | 0.603    | 0.013   | 0.708      | 0.666    |
| spatially res.   | structured | short        | 0.687      | 0.633    | 0.017   | 0.752      | 0.724    |
| spatially res.   | naive      | short        | 0.653      | 0.605    | 0.004   | 0.739      | 0.709    |
| aggregated       | -          | short        | 0.485      | 0.318    | 0.126   | 0.610      | 0.513    |

**Table A1.** Various performance metrics for the experiments presented in section 5. The metrics were calculated after picking the best out of ten random seeds for each condition in terms of median NSE.

| Input Processing | Routing    | initial LR | median NSE |
|------------------|------------|------------|------------|
| spatially res.   | structured | 5e-4       | 0.738      |
| spatially res.   | naive      | 5e-4       | 0.724      |
| aggregated       | -          | 5e-4       | 0.691      |
| spatially res.   | structured | 1e-3       | 0.745      |
| spatially res.   | naive      | 1e-3       | 0.721      |
| aggregated       | -          | 1e-3       | 0.704      |

**Table A2.** Median NSE over 20 seeds for exploratory experiments on the optimal initial learning rate. Performance was evaluated on the *validation* period, as this is part of the model selection process.

Author contributions. M.A.V. and J.M. designed the experiments with crucial suggestions from N.O. M.A.V. prepared the data, implemented the model, performed experiments, and analyzed the results. M.A.V. prepared the manuscript with significant contributions from J.M and N.O. All authors reviewed the manuscript.

Competing interests. The authors declare no competing interests.

Acknowledgements. This work was supported by the Federal Ministry for Economic Affairs and Climate Action (BMWK) as grant DAKI-FWS (01MK21009A).

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
