# Peer review of "Spatially Resolved Rainfall Streamflow Modeling in Central Europe"

_EGUsphere, 2024_

## Author Response (AR1)

Dear Editor and Reviewers,

we thank you all very kindly for the constructive comments, helpful feedback and time spent in reviewing the manuscript. This process has increased the quality of the manuscript substantially, and we hope you are also satisfied with it. In this document we provide a point-by-point response to both reviews, followed by a list of changes to the manuscript.

**Review 1**

• R1.1. While the proposed pipeline is compared with its own simplified variants (e.g., aggregated processing and naive routing), the manuscript does not include any direct comparison with established hydrological or deep learning models, such as traditional LSTM-based models or conceptual hydrological models. This limits the ability to assess the relative performance and novelty of the proposed approach. Moreover, the results section lacks standard comparative elements such as performance tables, or time series comparisons that would help illustrate how the model performs relative to well-known baselines. Including such benchmarks is important to validate its practical advantages.

A1.1. We appreciate the review pointing this out. Indeed, our aggregated baseline is the established LSTM model introduced by Kratzert (2019), apart from a few small difference in parameter values. Since its introduction, this model has been used in a number of studies across different regions and has become the benchmark neural network model. We changed the manuscript to make this point more clear to the reader in the Subsection "Baselines".

**4.5 Baselines**

We introduce two baselines in order to evaluate the performance gains of the two central aspects of our pipeline: spatially resolved processing in the local stage and the inclusion of structural bias in the routing stage. In the first baseline experiment, we aggregate all spatial information within a catchment and feed it through the same architecture as used in the local stage. Normally, this stage of the model processes individual grid cells. But here it processes entire, aggregated catchments, and the output is a prediction of the runoff measured at the corresponding station. The aggregated baseline does not require any further routing. This model and processing pipeline is identical to the one in Kratzert et al. (2019b), apart from the small differences in parameter values listed in Subsection 4.1. As we discussed above, it has been widely used in the literature since its introduction, which allows us to compare our own results from our custom dataset to a wider body of literature. We will call this baseline aggregated, whereas our default model will be referred to as spatially resolved.

We also made sure to explicitly compare the results of our baseline to the results achieved by the same model in other studies. To this end, we included a new paragraph right at the beginning of "Results and Discussion" before discussing our own model.

**5.1 Model Performance**

The aggregated baseline achieves a median NSE test performance of 0.69 on our dataset. As discussed above, it is virtually identical to the model from Kratzert et al. (2019b), who report a median NSE of 0.74 on the CAMELS dataset. These results were subsequently confirmed, small modifications of the model parameters or test period notwithstanding, by Shalev et al. (2019) (median NSE 0.73), Acuña Espinoza et al. (2024) (median NSE 0.75) and Frame et al. (2022) (median NSE between 0.72 and 0.81) We conjecture that our baseline performance is probably lower due to our dataset being smaller than CAMELS and likely containing greater human activity. Turning to other regions, the same model was used for example by Mai et al. (2022) for the Great Lakes Area (median KGE of 0.76), by Lees et al. (2021) for Great Britain (median NSE 0.88) and by Loritz et al. (2024) for Germany (median NSE 0.84). The latter two results go against our conjecture for inferior performance due to human signal. The fact that Loritz et al. (2024) use an ensemble of models might be a factor at play, but in any case this requires further investigation. These studies provide valuable comparisons with other model types: In all studies just mentioned, the authors compared the neural network model against a variety of physical, conceptual or hybrid models and consistently found that the alternatives were outperformed by the neural network.

As far as physical or conceptual models are concerned, our dataset does not contain features required for the regional calibration that is usually done with these models. We would very much like to see a proper comparison of our model to a physical/conceptual model, but the dataset would need to be extended first (particularly with static hydrological signatures like in the CAMELS dataset). This was beyond the scope of this paper, but we are currently working on re-creating a spatially resolved version of the CAMELS dataset to provide more straightforward comparability

in the future. Again, this was unfortunately beyond the scope of the present paper. Finally, we also discuss that the referenced studies employing the baseline model in turn contain comparisons with physical and conceptual models on their respective datasets. In all these studies, the baseline LSTM model consistently outperformed the other model types. Your point about performance measures is also very valid: We included a summary version of appendix A2 with the most important experimental conditions and performance metrics into the main text. This way, the reader can get a better initial idea about the model performance without having to jump to the appendix.

| Input Processing   | Routing    | median NSE | NSE

Figure 1. Overview of study area, input grid and data types. (a) The study area comprises 5 basins that cover a contiguous area in central Europe2. (b) Cells of input grid (orange) for Upper Danube basin. Catchment boundaries (black) are overlaid with corresponding stations (blue), as well as connecting arrows representing the station connectivity network. Cells along catchment boundaries are assigned entirely to the catchment that contains their center point. (c) Visualizations for one example feature of each type of input. Basin outlines (black) and borders of Germany (turquoise) are plotted for reference.

**2.3 Contributions**

In this paper, we present three contributions that go beyond what we have discussed so far: We process the data in a spatially resolved manner without prior aggregation, we use a simple routing module that allows for interpretability, and we train this model in an end-to-end fashion on a novel, spatially resolved dataset in central Europe.

Spatially resolved processing takes place in the first or *local* stage of our model, detailed in Subsection 4.1. Its architecture largely follows the one presented in Kratzert et al. (2019b). Instead of using lumped basins as inputs, we apply the same neural network in parallel to each cell of a regular grid of meteorological time series and ancillary inputs. We show that this finer spatial resolution allows us to capture co-variances and provide regularization, especially benefiting larger catchments. This approach is natural for physical or conceptual models that solve local equations at a given rasterization. Yet, no one to our knowledge has applied a neural network directly to the grid of inputs in a way that scales up to entire river basins. The local stage yields a local runoff quantity for each grid cell, exemplarily visualized in the right panel of Figure 6.

The second or *routing* stage, detailed in Subsection 4.2, consists of only two simple network layers without any nonlinearity, efficiently mapping these local runoff quantities onto a river network. Both stages are trained jointly in an end-to-end fashion on the entire dataset, rendering any kind of expert knowledge obsolete. This also means that the model is fitted exclusively on empirical data, enabling scientific discovery from raw data. We show that the river network connectivity graph can be used as inductive bias to constrain the model to reproduce the river's natural layout. This increases data efficiency and allows for better interpretability. We explain how in principle, although this has yet to be shown in practice, the model can be controlled interactively: Extracting or injecting quantities of water can simulate human influence such as industrial, agricultural or hydroelectric energy generation activity, which significantly contributes to streamflow but is independent of the modeled hydrological processes.

Lastly, the lumped datasets discussed above are unsuited for this spatially resolved modeling approach, since it requires both non-spatially-aggregated inputs and streamflow data for entire basins. Hence, for this study we compiled gridded meteorological and static data as well as river streamflow records for five entire basins in central Europe. These basins are characterized by an overall high level of human activity, compared to the CAMELS dataset. The data is publicly available and described in more detail in Vischer et al. (2025, under review).

• R1.4. The authors mention that they include sine-cosine embeddings of the day of the week and the day of the year, describing them as a coarse proxy for human activity. However, this design choice is not clearly linked to the later discussion in the results section on human influence. It is unclear how these embeddings contribute to modeling human activity or whether they have any measurable effect on model performance.

A1.4. In order to address the question of human activity, some datasets like e.g. GAGES-II include static estimates of human activity within each catchment. These are usually based on map data such as e.g. roads or population density. We did include a map of land use, but thoroughly deriving such estimates was unfortunately beyond the scope of this paper. Similarly for temporal aspects of

human activity, to properly evaluate the sine-cosine embedding, we would need some time series data or estimates of human activity for the study period, which are even harder to come by. We want to investigate this important question in a principled manner in a context where we have suitable ground truth data. For the purpose of this paper, we decided to focus on developing a suitable model structure as a first step.

• R1.5. It appears that the input time window used in the local stage is fixed at nine days. The paper does not clearly explain how this value was selected. And there is no discussion of how the model performs under different forecast horizons. This is a critical aspect for practical forecasting applications.

A1.5. The 9 days parameter is actually the size of the time-convolution kernel in the routing module, meaning that the routing has a "memory" of 9 days. We determined this by a back-ofenvelope calculation of the maximum time that water would flow inside the river networks that we investigated and multiplied the result by 2 as a generous safety margin. The input length to our model can in fact be chosen flexibly: The LSTM and routing modules do not map sequences of fixed lengths, but instead allow for continuous mapping, one day at a time. For the study, we used an input length of 400 days (with varying start days of year) to fully capture the yearly hydrological cycle plus some safety margin. For inference, the entire length of validation and test sets (several years) are processed in one sweep. As you point out correctly, the forecast horizon is extremely relevant for practical applications. Please note, however, that our model relies on the weather forecast as input and integrates it with its own representation of the state of the system in order to generate predictions for the future. The performance of our model thus hinges on the accuracy of the weather forecast. For the purpose of this study, we thus limited ourselves to reanalysis data as a first step in order to obtain clear results as far as our model is concerned. We agree that before using our model or the results of this paper in real world operations, a thorough evaluation of the accuracy relative to the forecast horizon should be performed, comparing various weather models and perhaps and ensemble of such models. In practice, we would assume the quality of the weather forecast and thus the quality of our model to be good for a few days, acceptable for a week and deteriorating over another week before becoming practically unusable.

**Review 2**

Figure 1. Overview of study area, input grid and data types. (a) The study area comprises 5 basins that cover a contiguous area in central Europe2. (b) Cells of input grid (orange) for Upper Danube basin. Catchment boundaries (black) are overlaid with corresponding stations (blue), as well as connecting arrows representing the station connectivity network. Cells along catchment boundaries are assigned entirely to the catchment that contains their center point. (c) Visualizations for one example feature of each type of input. Basin outlines (black) and borders of Germany (turquoise) are plotted for reference.

• R2.1. The manuscript is dense and could benefit from clearer section transitions and subheadings to improve readability.

- A2.1. Thank you very much for this comment. We have substantially modified the manuscript and believe that these changes make it now clearer and more readable. In order to make all changes readily noticeable, we attached a latex differential file (author's track-changes file) that highlights the changes right next to the original version. Following the suggestion we separated the Section "Data and Methods" into two separate sections "Data" and "Methods".
- R2.2. The study area map lacks coordinates, which suggests that the authors wrote the paper carelessly. Kindly insert the coordinates.
  - A2.2. We inserted the coordinates and combined your other figure-related suggestions into a new, paneled data overview figure that you can see at the beginning of this section. More on this below.
- R2.3. Please use DEM 30 m or 90 m resolution to prepare the study area map.
  - A2.3. Thank you for the suggestion. We used the Copernicus DEM 90 m to create the map of the study area featured in the data overview figure above.
- R2.4. Show the study area map on a world map. This will be very helpful for international readers. A2.4. Thank you for the suggestion. We decided to display the study area on a map of Europe instead of a global world map to be more precise and readable.
- R2.5. The methods section should not include the study area portion. Please create a separate section for the study area. Other significant information regarding the study region should also be included in the study area section using figures and graphs.
  - A2.5. The new "Data" section now features a "Study Area" subsection with a paneled figure which contains the updated input type maps (discussed above), a map of the study area in Europe (discussed above) and a visualization of the input grid that the first reviewer requested. You can see the combined figure at the beginning of this section. We hope that this provides the reader with a clear yet comprehensive visual explanation of the study area and input data pipeline.
- R2.6. Figures are informative but would benefit from more transparent labels and captions.
   A2.6. We improved the figure labels and captions. Please compare the changes in the abovementioned track-changes file.
- R2.7. Some sentences are overly long or awkwardly phrased. A thorough proofreading for clarity and conciseness is recommended.
  - A2.7. We proofread the entire manuscript and made a number of changes to improve readability. Please refer to the track-changes file.

**List of Changes to the Manuscript**

- We changed Subsections "Baselines" and "Model Performance" to make more explicit that our
  baseline model is the model used in the wider literature, and compare our results more explicitly.
  We created a summary version of the table comparing performance metrics and added it in the
  beginning of the Section "Results and Discussion".
- We fixed a bug in the KGE metrics in Appendix Table A2.
- We created a new data overview figure that contains an overview map of the study area, a visualization of the input grid and station network as well as an overview of the types of input data.
- In creating this figure, we followed the second reviewer's advice regarding coordinates, using the Copernicus DEM and plotting the study area on a larger map for context.
- We added a clarifying sentence on how the grid cells are handled to our description of the input grid.
- We re-wrote the Subsection "Contributions" to be more clearly structured and more readily understandable.
- We split up the Section "Data and Methods" into two separate sections "Data" and "Methods". We added a dedicated Subsection "Study Area" to the new Section "Data".
- We re-formulated some figure captions in order to make them more clear and easily understandable.
- We made a number of minor changes to the manuscript text, particularly to the section transitions, in order to improve on the conciseness and legibility of the manuscript.